# A Study of the Dielectric Relaxation of Nitrile–Butadiene Rubber, Ethylene–Propylene–Diene Monomer, and Fluoroelastomer Polymers with a Self-Developed Deconvolution Analysis Program

**DOI:** 10.3390/polym17111539

**Published:** 2025-05-31

**Authors:** Youngil Moon, Gyunghyun Kim, Jaekap Jung

**Affiliations:** 1Department of Electrical Engineering, Pohang University of Science and Technology, Pohang 37673, Gyeonsangbuk-do, Republic of Korea; yimoon@postech.ac.kr; 2Advanced Institute of Convergence Technology, Suwon 16229, Gyeonggi-do, Republic of Korea; gyunghyun_eeee@snu.ac.kr; 3Hydrogen Energy Materials Research Center, Korea Research Institute of Standards and Science, Daejeon 34113, Republic of Korea

**Keywords:** dielectric relaxation, impedance spectroscopy, activation energy, glass transition temperature, nitrile–butadiene rubber, ethylene–propylene–diene monomer, fluoroelastomer

## Abstract

This study presents an integrated analysis of the dielectric characteristics of nitrile–butadiene rubber (NBR), ethylene–propylene–diene monomer (EPDM), and fluoroelastomer (FKM) polymers. Dispersion spectra were obtained over a wide range of frequencies and temperatures, and, via our self-developed “Dispersion Analysis” program, the obtained dielectric spectra were precisely deconvoluted. Notably, α, α’, β, and γ relaxation phenomena, including the DC conduction process, were identified in NBR, whereas three relaxation processes, namely, α, β, and the Maxwell‒Wagner‒Sillars (MWS) process, as well as DC conduction, were observed in EPDM and FKM copolymers. The activation energies ($E_{a}$) for secondary relaxation—namely, β, γ, and MWS—and the DC conduction process, which are observed in NBR, EPDM, and FKM, were determined via the Arrhenius temperature dependence model, and these values were compared with previously published results. Furthermore, the glass transition temperature ($T_{g}$), extrapolated from the relaxation rate of the α process, was estimated via the Vogel–Fulcher–Tamman–Hesse (VFTH) law. The values of $T_{g}$ obtained using dielectric spectroscopy for NBR, EPDM, and FKM agreed well with the differential scanning calorimetry (DSC) measurements. This study provides foundational insights into the dielectric properties of widely used rubber polymers, offering a comprehensive reference for future research.

## 1. Introduction

Owing to their outstanding physical, chemical, and mechanical features, polymeric rubber compounds have been widely utilized in various industrial applications, such as O-ring gaskets, pipelines and valve sealants, electrical insulation tubes, and automotive and aerospace components [1,2,3,4,5]. Recently, rubber compounds have played a crucial role in hydrogen electric vehicle (HEV) applications by increasing the load-bearing capacity, reducing rolling resistance, and improving durability [6,7,8,9]. Furthermore, low chemical reactivity and minimal outgassing properties are essential for hydrogen fuel cells and high-vacuum applications (down to approximately 1^−9^ Torr) [10,11,12,13,14,15,16,17,18,19,20]. Thus, extensive research has been conducted to develop advanced rubber composites, and various experimental techniques have been employed to evaluate their physical and mechanical characteristics [21,22,23,24,25,26,27,28,29,30,31,32,33,34,35,36,37,38,39,40,41,42,43,44,45,46,47,48,49,50,51,52,53,54,55,56,57,58,59,60,61,62,63,64,65,66,67]. Representative methods—such as tensile testing, dynamic mechanical analysis (DMA), transport analysis, thermogravimetric analysis (TGA), Fourier-transform infrared spectroscopy (FTIR), nuclear magnetic resonance (NMR), and energy-dispersive X-ray spectroscopy (EDS)—have provided valuable insights into the structural and functional attributes of these materials [68,69,70,71,72,73,74,75,76,77,78,79]. Among these techniques, dielectric spectroscopy stands out as a nondestructive and effective method for evaluating the electrical characteristics of rubber composites as it enables precise measurements to be made of polarization dynamics and charge carrier transport phenomena across a broad frequency range of alternating electric fields.

Debye, who first illuminated the molecular aspects of dielectric behavior, established the foundational concepts and models underlying dielectric spectroscopy [80,81,82]. This technique offers unique advantages, including the ability to probe molecular dynamics, charge carrier distributions, and determine a material’s response to external electric fields ranging from direct current (DC) to approximately 10^12^ Hz [83,84,85,86,87,88,89,90,91,92,93,94,95]. It is particularly useful for understanding the behavior of polarized molecular structures under various environmental conditions, including temperature and humidity variations [96,97,98,99,100,101,102,103,104,105,106,107,108]. Consequently, numerous studies have employed dielectric spectroscopy to investigate the dielectric properties of various polymeric rubbers, such as styrene−butadiene rubber (SBR), nitrile–butadiene rubber (NBR), ethylene–propylene–diene monomer (EPDM), and fluoroelastomer (FKM) [74,109,110,111,112,113,114,115,116,117,118,119,120,121,122,123,124,125,126,127,128,129,130,131,132,133,134,135,136,137,138,139,140,141,142,143,144,145,146,147,148,149,150,151,152,153,154,155,156,157,158,159,160,161,162,163,164,165,166,167].

In our previous work, we reported comprehensive studies of NBR, EPDM, and FKM rubbers via dielectric spectroscopy over a wide range of frequencies and temperatures [33,168,169,170,171,172,173]. In the present study, we review the dielectric analysis results of these rubber composites via a self-developed deconvolution analysis program. By deconvoluting the obtained dielectric spectra with a combination of the Havriliak–Negami model and an exponential power law for DC conduction, we identified various relaxation phenomena, including the DC conduction process. The origins of each relaxation were determined on the basis of their temperature-dependent relaxation times and the corresponding dipole orientation dynamics of specific molecular structures. Additionally, the activation energies for the relaxation processes and DC conduction, both of which follow the Arrhenius temperature dependence law, were obtained and compared with previously reported values. Moreover, the glass transition temperature ($T_{g}$) was calculated from the α relaxation, which follows the Vogel–Fulcher–Tammann–Hesse (VFTH) temperature dependence. The obtained $T_{g}$ value was then compared with the value obtained via differential scanning calorimetry (DSC). Furthermore, the basic algorithm and outlook of the self-developed dielectric deconvolution program based on C# were introduced. The entire code of this developed deconvolution software is available in the Appendix A. This study provides fundamental insights into the dielectric properties of widely used rubber polymers and offers a comprehensive reference for future research.

## 2. Relaxation Process and Model Analysis

Although numerous relaxation and transition dynamics have been studied across various types of materials [93,174,175,176,177,178,179,180,181,182,183,184,185,186,187], we briefly discuss the relaxation processes limited to synthetic rubber polymers. The internal structure of these polymers primarily comprises repeated monomer chains, filling, and vulcanizing agents [188,189,190]. In the millimeter to microwave frequency range, the dielectric response of rubber polymers is mainly attributed to the dipole orientation of the backbone and side chains as well as interfacial polarization processes [191,192,193,194,195,196,197,198,199,200]. Each of these relaxation processes can be characterized by evaluating the shape of the dispersion spectra and their temperature dependence. The underlying mechanisms of these dielectric relaxation processes, along with the corresponding analytical functions, are given below.

### 2.1. α Relaxation

The α relaxation process represents the dominant relaxation behavior in amorphous polymers, originating from the reorientation of segmental (backbone) chain motions [199,200,201]. It is directly associated with the polymer’s glass transition dynamics and is thus often referred to as the dynamic glass transition. The relaxation rate ($f_{p,\alpha }$) of α relaxation obeys the non-Arrhenius temperature dependence described by the VFTH temperature dependence model:(1)$$\ln f_{p,\alpha }=\ln f_{p,\infty }-\frac{A}{T-T_{0}}$$
where $f_{p,\infty }$ and A are constants; $T_{0}$ is the ideal glass transition temperature or Vogel temperature; and the static $T_{g}$, determined using dielectric spectroscopy, corresponds to the temperature at which the curve of the VFTH temperature behavior is attained at approximately 1.6 mHz [199]. This frequency can vary depending on the thermal protocol. Meanwhile, the dynamic $T_{g}$ is identified as the temperature showing the peak in dielectric loss at a specified measurement frequency.

### 2.2. α′ Relaxation (Normal Mode Relaxation)

The α′ relaxation process is associated with fluctuations in the end-to-end vector of polymer chains and is characteristic of type A polymers. According to Stockmayer [202] and Block [203], the vector sum of the total dipole moment in polymer chains comprises the parallel direction with the segmental chain (backbone chain) called polymer type A. α′ relaxation, called normal mode relaxation, can be observed when it is proportional to the fluctuation of the end-to-end vector of the polymer chain. Like α relaxation, α′ relaxation has a central frequency with a nonlinear temperature dependence. The relaxation rate of α′ can also be described by VFTH temperature dependence.

### 2.3. β and γ Relaxations (Secondary Relaxation)

β and γ relaxations are a type of secondary relaxation resulting from the fluctuating motion of the side groups attached to polymer chains [199,200,201]. In addition, secondary relaxations may also originate from isolated molecules or impurities, which exhibit high relaxation rates. The activation energy ($E_{a}$) for these relaxations is determined by assuming an Arrhenius temperature dependence:(2)$$f_{p}=f_{\infty }\exp\left[-\frac{E_{a}}{k_{B}T}\right]$$
where $f_{p}$ is the position of the center peak of the secondary relaxation rate, $f_{\infty }$ is the pre-exponential factor, and $E_{a}$ is related to the slope of log($f_{p}$) versus 1/*T* and depends on the internal rotation barriers and the environment of a moving unit.

### 2.4. Interfacial Relaxation (MWS Relaxation)

Interfacial relaxation, called Maxwell‒Wagner–Sillars (MWS) relaxation, originates from polarization processes between polymer chains and filler contents or other elements in a heterogeneous polymeric system [199,200,201]. These relaxation phenomena can only be detected via dielectric spectroscopy. Thus, dielectric measurements can provide valuable information on the morphology of a phase-separated polymer system. Similarly to secondary relaxation, interfacial relaxation has a relaxation rate on interfacial polarization with an Arrhenius temperature dependence determined via Equation (2).

### 2.5. DC Conduction Process

The contribution of the DC conduction process to dielectric spectra typically appears in the low-frequency (a few millihertz to kilohertz) and high-temperature (above 280~300 K) regimes of the dielectric loss spectrum. The origin of this process in polymers can be regarded as a transport phenomenon by considering the movement of charge carriers such as ionic impurities [204] and thermally excited electrons between the valence and conduction bands [205]. The complex dielectric constant $\varepsilon^{*}\left(\omega \right)$ is written as [199,200,201](3)$$\varepsilon^{*}\left(\omega \right)=\varepsilon^{\prime }\left(\omega \right)-\varepsilon ”(\omega )=\varepsilon^{\prime }\left(\omega \right)-i\frac{\sigma (\omega )}{\varepsilon_{0}\omega }$$
where σ(ω) is the total conductivity, and $\varepsilon_{0}$ is the vacuum permittivity of 8.854 × 10^−12^ F/m. The measured dielectric loss $\frac{\sigma (\omega )}{\varepsilon_{0}\omega }$ is composed of two components, namely, one due to dielectric loss ε″(ω) and one due to DC electrical conductivity $\sigma_{dc}$, which is the low-frequency limit of σ(ω). Thus, the general expression of Equation (3) is written as(4)$$\varepsilon^{*}\left(\omega \right)=\varepsilon^{\prime }\left(\omega \right)-i\left[\varepsilon \left(\omega \right)+\frac{\sigma_{dc}}{\varepsilon_{0}\omega }\right]$$

The DC conductivity loss, $\frac{\sigma_{dc}}{\varepsilon_{0}\omega }$, increases with decreasing frequency. The temperature dependence of $\sigma_{dc}$ can be described by the VFTH model:(5)$$\sigma_{dc}=\sigma_{0}\exp\left[-\frac{E_{a}}{k_{B}(T-T_{0})}\right]$$
where $\sigma_{0}$ is the maximum conductivity at the temperature of infinity and is proportional to the number of charge carriers. $k_{B}$ is Boltzmann’s constant. $T_{0}$ is the Vogel temperature, which defines the low-temperature limit of the conductivity curves and is the temperature at which the ion mobility approaches zero. $E_{a}$ is the activation energy of the conduction process. In some systems, especially those without strong glassy dynamics, the temperature dependence of $\sigma_{dc}$ is better described by the Arrhenius law. This behavior reflects thermally activated hopping transport through a more uniform energy landscape [199].

### 2.6. Models for Analyzing the Relaxation Process

The permittivity of dielectric materials varies with the frequency ω of an alternating electric field. The permittivity is normally expressed by the complex permittivity ε* of a medium as a function of the angular frequency, ω = 2πf. The simplest model is the Debye dielectric model [80], which is written as follows:(6)$$\varepsilon^{*}\left(\omega \right)=\varepsilon_{\infty }+\frac{\varepsilon_{s}-\varepsilon_{\infty }}{1+i\omega \tau_{0}}$$
where $\varepsilon_{\infty }$ is the real permittivity at the high-frequency limit (ω = ∞). $\varepsilon_{s}$ is the real permittivity under a static electric field (ω = 0). $\tau_{0}$ is the characteristic relaxation time for the polarization to reach an equilibrium state after a disturbance. $\varepsilon_{s}-\varepsilon_{\infty }$ is the dielectric relaxation strength of the corresponding relaxation loss peak.

In the case of relaxation with respect to time distribution, the Debye dielectric relaxation model was expanded to the Cole‒Cole model [206] by introducing the empirical parameter α:(7)$$\varepsilon^{*}\left(\omega \right)=\varepsilon_{\infty }+\frac{\varepsilon_{s}-\varepsilon_{\infty }}{1+{\left(i\omega \tau_{0}\right)}^{1-\alpha }}$$

This Cole‒Cole equation is used when the dielectric loss peak shows symmetric broadening. α lies in the range of 0 ≤ α ≤ 1. For α = 0, the Cole‒Cole model becomes the Debye model. Furthermore, the Cole‒Davidson model [207] with a different empirical parameter β was modified as follows:(8)$$\varepsilon^{*}\left(\omega \right)=\varepsilon_{\infty }+\frac{\varepsilon_{s}-\varepsilon_{\infty }}{{\left(1+i\omega \tau_{0}\right)}^{\beta }}$$

The two expanded models were integrated into the Havriliak‒Negami (HN) model [208] by introducing the two empirical parameters, α and β, as follows:(9)$$\varepsilon^{*}\left(\omega \right)=\varepsilon_{\infty }+\frac{\varepsilon_{s}-\varepsilon_{\infty }}{{\left(1+{\left(i\omega \tau_{0}\right)}^{1-\alpha }\right)}^{\beta }}$$

This equation considers symmetric and asymmetric broadening. α and β are shape parameters with values between 0 and 1 that depend on the relaxation time distribution. α and β are related to the linewidth and asymmetric relaxation loss peak, respectively. As α and β approach zero, the relaxation spectrum exhibits increased broadening and a more asymmetrical shape. To understand the dynamics of the relaxation processes in the polymer, the data for the experimental complex dielectric permittivity, $\varepsilon^{*}\left(\omega \right)$, were fitted with the HN function. The HN function formalism can be phenomenologically described as a combination of the conductivity term with the HN functional form as follows [199,200,201]:(10)$$\varepsilon^{*}\left(\omega \right)=-i{\left(\frac{\sigma_{dc}}{\varepsilon_{0}\omega }\right)}^{N}+\varepsilon_{\infty }+\sum_{k=1,2\ldots }\frac{∆\varepsilon_{k}}{{\left(1+{\left(i\omega \tau_{HN_{k}}\right)}^{1-\alpha_{k}}\right)}^{\beta_{k}}}$$
where the first term accounts for conductivity. *N* is an exponent that characterizes the nature of the conduction process as well as the electrode polarization effect on the dielectric dispersion spectrum. This contribution becomes most significant at high temperatures and low frequencies. The temperature dependence of the exponent N for EPDM and FKM is presented in Appendix A. The last term is the HN function. The summation symbol indicates the occurrence of more than one relaxation process. $\tau_{HN_{k}}$ is the average characteristic relaxation time of the corresponding process *k*. The exponents $\alpha_{k}$ and $\beta_{k}$ (0 ≤ $\alpha_{k}$ ≤ 1, $\beta_{k}$ ≤ 1) are the HN fitting parameters (shape parameters) describing the distributions of the relaxation times. The dielectric relaxation strength, $\varepsilon_{s}-\varepsilon_{\infty }$, is represented by $∆\varepsilon_{k}$.

## 3. Methods

### 3.1. Dielectric Spectroscopy

The dielectric spectroscopy system (Figure 1) used to measure the complex permittivity comprises a temperature-controlled chamber, a digital multimeter (Fluke 8842A) for temperature monitoring, and an impedance analyzer (VSP-300, Bio Logics) with a GPIB interface to a PC. The temperature is controlled and maintained by proportional–integral–derivative (PID) control with a heater and a circulating refrigerating medium, varying at intervals of 5 K with a temperature stability better than 0.1 K at temperatures ranging from 233 to 404 K. Cylindrical samples (50 mm in diameter and 2.4 mm in thickness) are placed between two circle-shaped copper electrodes. An AC voltage is applied to the electrodes, the resulting current is measured, and the impedance is obtained. The measured complex impedance values ($Z^{\prime }$ and $Z^{\prime \prime }$) are then converted to complex permittivity values via the following relationships:(11)$$\varepsilon^{\prime }=\frac{d}{\omega \varepsilon_{0}A}\frac{Z^{\prime \prime }}{{Z^{\prime }}^{2}+{Z^{\prime \prime }}^{2}},\;\varepsilon^{\prime \prime }=\frac{d}{\omega \varepsilon_{0}A}\frac{Z^{\prime }}{{Z^{\prime }}^{2}+{Z^{\prime \prime }}^{2}},$$

### 3.2. Differential Scanning Calorimetry

Differential scanning calorimetry (DSC) was employed to determine the dynamic glass transition temperature using a Q-600 instrument (TA Instruments, New Castle, DE, USA). Approximately 10 mg of granulated rubber samples was analyzed at a controlled heating rate of 1 °C/min over the temperature range of −80 °C to 80 °C.

## 4. Development of Self-Developed Deconvolution Analysis Program

In numerous previous studies, deconvolution analysis via a combination of Debye-based fitting models was performed with the help of numerical fitting software or algorithms developed in C#, Python, and MATLAB. Additionally, commercialized and open-access or publicly available graphical user interface (GUI) software for data analysis also exists, as shown in Figure 2a,b. These commercial software packages offer various useful features, including support for nonlinear curve fitting via the Havriliak‒Negami model. For example, the “EIS-smart-tool” (Figure 2a) provides Nyquist plots for equivalent circuit analysis and an interactive GUI that allows for the easy rotation of 3D plots, along with a convenient fitting setup for Arrhenius plots. Furthermore, the “WinFit” (Figure 2b) software supports a diverse range of domain analysis functions, including frequency, temperature, and time. Notably, the temperature and time domain analysis features, which are rarely found in other impedance analysis software, enable the easy determination of the activation energies and glass transition temperatures.

However, in practical experimental settings, impedance data are acquired across numerous temperature points to track the evolution of dielectric relaxations with thermal variation. As a result, these tools require repetitive manual fitting procedures for each dataset, significantly increasing the analysis time and reducing overall efficiency.

To address this challenge, we developed a custom data analysis software named “Dispersion Analyzer” to facilitate comprehensive and consistent deconvolution across wide temperature ranges. The software incorporates a smoothness constraint function that enables the simultaneous fitting of multiple datasets while maintaining physical continuity between temperature-dependent parameters. Unlike conventional tools that fit each temperature individually, our software supports the following:Weighted fitting between the real and imaginary parts of permittivity spectra to emphasize dominant spectral contributions;Global fitting across temperature series under physically meaningful constraints (e.g., shared activation energy or Arrhenius/VFT behavior);User-controllable fitting accuracy achieved by adjusting convergence thresholds, iteration limits, and optimization tolerances.

These capabilities allow for both interpretability and efficiency in complex dielectric analysis, which—based on our assessment—are not commonly supported in existing commercial packages.

The developed software and additional information are provided in the Appendix A.

The self-developed deconvolution analysis program, which was written in C# using Microsoft Visual Studio Net 2018, has the GUI shown in Figure 3. It can be executed on a Microsoft Windows system. The dimensional information of the sample can be inserted into the material information at the bottom left of the main panel (presented in Figure 3) to extract the complex permittivity value from the raw impedance data. The software provides two-dimensional plots on the right window and three-dimensional plots on the left window to visualize the complex permittivity spectra as a function of frequency and temperature. Moreover, users can easily configure the fitting information, such as the number of fitting functions, constraints on fitting parameters, and weighting for the variance of fitting parameters across closely related temperatures. In the three-dimensional plot on the left window, the red and blue surfaces represent the real and imaginary parts of the complex permittivity, respectively, and the solid line represents the dielectric spectra at the specific temperature, which is displayed in the two-dimensional plot in the right panel of the main window. Furthermore, the red and blue cross symbols in two-dimensional plots indicate the measured real and imaginary permittivity data values, respectively, whereas the solid lines represent the fitted results obtained via a combination of two HN functions. The ten parameters obtained via the two HN model functions in model information are listed in the lower right section of the main window.

## 5. Results and Discussion

### 5.1. NBR

NBR is a synthetic copolymer composed of butadiene (CH_2_CH=CHCH_2_) and acrylonitrile (CH_2_CHCN). Its representative molecular structure is illustrated in Figure 4 [169]. The detailed chemical compositions of the NBR samples used in this study are summarized in Table 1.

The three-dimensional frequency and temperature dependences of the imaginary parts of the permittivity of NBR are shown in Figure 5a. All four relaxation processes, called α, α’, β, and γ relaxations, and the contribution of the DC conduction process were observed in the experimental temperature range (233–404 K) and frequency range (0.01–3 MHz). The yellow arrows in Figure 5a indicate the evolution of the relaxation peaks versus the temperature variation. The temperature ranges observed for each relaxation process are presented (Figure 5b) as follows:(i).The temperature range for β relaxation: 233–269 K.(ii).The temperature range for γ relaxation: 233–269 K.(iii).The temperature range for α relaxation: 233–341 K.(iv).The temperature range for α’ relaxation: 254–404 K.(v).The temperature range for DC conduction contribution: 324–404 K.

The simulation results of the relaxation processes for seven representative temperatures (404 K, 359 K, 326 K, 293 K, 272 K, 251 K, and 233 K) in the frequency domain after deconvolution via the self-developed deconvolution analysis program are shown in Figure 6a–g. The α′ and α relaxation processes were well described by the HN function, as given in Equation (10). The β and γ relaxations, exhibiting symmetrical loss profiles, were fitted using the Cole–Cole equation, which corresponds to the case of *β* = 1 in the HN model. As the temperature decreased, all relaxation loss spectra shifted toward lower frequencies.

The α’ relaxation, known as normal mode relaxation, reflects fluctuations in the end-to-end dipole vector of a polymer chain. This phenomenon typically occurs in Stockmayer type A polymers, where the dipole moment aligns along the polymer backbone [202,209,210,211]. Among the studied materials, this α’ relaxation was identified exclusively in NBR due to its dipole vector being oriented parallel to the chain backbone. In contrast, the α relaxation is linked to the cooperative segmental dynamics of the polymer backbone itself [212,213,214,215]. The β and γ relaxations, on the other hand, originated from the orientational movements of side groups—vinyl (–C₂H₃) and cyano (–CN), respectively. Notably, the γ relaxation, which involves the motion of the smaller cyano groups, typically appears at higher frequencies or lower temperatures than the β relaxation in isochronal measurements [201,216,217,218]. Figure 7 provides a schematic of the NBR molecular structure along with its characteristic relaxation processes.

Figure 8 presents the center frequency (f₀) of the relaxation peak for NBR, plotted as a function of the reciprocal temperature. As previously observed in Figure 6, the relaxation peak shifts to lower frequencies as the temperature decreases. The activation energies for the β, γ, and DC relaxation processes were extracted using the Arrhenius relationship, as expressed in Equation (2). This analysis showed that the β and γ processes exhibited activation energies of 32.0 kJ/mol and 24.3 kJ/mol, respectively. Additionally, as the contribution of the DC conduction process increased with an increasing temperature, its activation energy was 38.7 kJ/mol. Pietrasik et al. [148] reported an activation energy of 32 kJ/mol for β relaxation; this value is identical to our experimental result. In contrast, Khalaf et al. [145] reported an activation energy of 22.41 kJ/mol for the DC conduction process, which is considerably different from our results. This discrepancy may be attributed to the differences in the type of carbon black (CB) filler used. In our study, we utilized medium thermal (MT) CB as a filling agent, whereas Khalaf et al. [145] used high-abrasion furnace (HAF) CB. HAF CB has a smaller primary particle size (30~40 nm) than MT CB (250~550 nm). Thus, the aggregation structure and percolation path in HAF-filled rubber composites are better formed than those in MT-filled composites. Consequently, the electrical conductivity of HAF-filled rubber is relatively high, resulting in relatively low activation energy. Similar results were reported in our previous study [173].

In Figure 6, the center positions of the loss peak for α show nonlinear temperature dependency; therefore, the fitting of α is conducted with the VFTH function. As mentioned in Figure 5, the α relaxation phenomenon results from the polarization of the backbone chain and is related to the glass transition mechanism. The $T_{g}$ value can be calculated by fitting it with the VFTH equation. The $T_{g}$ determined using dielectric relaxation spectroscopy (DRS) was identified as the temperature at which the extrapolated VFTH fit of the central frequency ($f_{0}$) versus 1000/T reaches 1.6 mHz, following an empirical criterion established in previous studies [199,200,201] (Figure 9a). Based on this method, the $T_{g}$ was determined to be 242.1 K (Figure 9a). In comparison, the $T_{g}$ obtained from DSC (Figure 9b) corresponds to the midpoint of the heat capacity change (1/2 ΔCp) often referred to as the “temperature of half-unfreezing” [201], yielding a value of 242.4 K. These two $T_{g}$ estimates show good agreement.

### 5.2. EPDM

EPDM is a synthetic rubber composed of ethylene (CH_2_CH_2_), propylene (CH_2_CHCH_3_), and ethylidene norbornene (ENB) units. Its representative molecular structure is illustrated in Figure 10, and the corresponding chemical composition is summarized in Table 2.

Figure 11a presents the three-dimensional frequency and temperature dependences of the imaginary part of the complex permittivity for EPDM. Within the experimental temperature range (233–404 K) and frequency range (0.01 Hz to 3 MHz), three distinct relaxation processes—α, β, and MWS relaxations—along with a conductivity contribution, were observed. Notably, a merged αβ relaxation behavior appeared in the temperature range of 305–404 K. The yellow arrows in Figure 11a indicate the evolution of the relaxation peaks with increasing temperature.

(i).The temperature range for α relaxation: 233 K to 302 K.(ii).The temperature range for β relaxation: 233 K to 305 K.(iii).The temperature range for αβ merged relaxation: 305 K to 404 K. (iv).The temperature range for the DC conduction contribution: 308 K to 404 K.(v).The temperature range for MWS relaxation: 233 K to 404 K.

The deconvoluted dielectric loss spectra of EPDM at seven representative temperatures (404 K, 359 K, 326 K, 293 K, 272 K, 251 K, and 233 K) are shown in Figure 12a–g, which show the evolution of three relaxation peaks and the DC conduction process with varying temperature. The black solid lines represent the total fitted curves, composed of individual components: MWS (orange dashed), β (blue dashed), and DC conductivity (navy dashed) fitted using two HN functions and a conductivity model for the spectra at 404 K, 359 K, 326 K, and 293 K or the fitted sums of the MWS (orange dashed line), α (red dashed line), and β (blue dashed line) relaxations by three HN fits (272 K, 251 K, and 233 K). The αβ merged relaxation process from 305 K to 404 K originates from the β process. All relaxation peaks shift toward higher frequencies with an increasing temperature.

The α relaxation in EPDM is observed at temperatures below room temperature. In the dielectric loss spectra, each frequency-dependent plot exhibits a broad and symmetrical loss peak, attributed to the local segmental motion of the polymer backbone. In contrast, the β relaxation, which appears across the entire temperature range studied, presents a narrower and symmetrical loss peak. This process is associated with the rotational dynamics of the side groups. Specifically, the α relaxation corresponds to the rotational motion around the main chain C–C bonds in the ethylene, propylene, and ENB units, as illustrated in Figure 13. Meanwhile, the β relaxation arises from the reorientation of the methyl (CH₃) groups located on the side chains of the propylene units, also depicted in Figure 13.

The position (f₀) of the central relaxation peak as a function of the reciprocal temperature for EPDM is presented in Figure 14. As the temperature decreases, the entire dielectric loss spectra shift toward lower frequencies. The inset of Figure 14 shows that the dielectric strength (Δε) of the α relaxation increases with decreasing temperature, while that of the β relaxation decreases. The temperature behavior of Δε is a general trend for α and β relaxation.

The activation energies for the β and MWS relaxation processes were estimated using the Arrhenius temperature dependence described by Equation (2), yielding values of 17.7 kJ/mol and 113 kJ/mol, respectively. Due to the limited availability of prior studies on the activation energies of EPDM rubber compounds, direct comparisons with the existing literature were not conducted in this work. As shown in Figure 15, the α and β relaxation peaks are clearly distinguishable in the temperature range of 233 K to 290 K, but gradually merge between 290 K and 373 K. The merging point at 290 K corresponds to the temperature where both processes exhibit the same relaxation time, as evidenced by the intersection in both the Δε (inset) and f₀ plots. This is the first time this has been observed in polymers with the help of the deconvolution analysis program. Additionally, because EPDM contains flexibly attached dipolar side groups (methyl groups (-CH_3_)), the value of Δε for β relaxation was smaller than that for α relaxation at low temperatures [219,220].

The temperature dependence of the α relaxation was analyzed using the VFTH model (Equation (1)). The T_g_ of EPDM was determined and compared with DSC measurements. The results obtained from both measurement methods are presented in Figure 15a,b. As shown in Figure 15a,b, the T_g_ value obtained from dielectric measurements was 227.3 K, while the DSC measurement yielded a value of 221.0 K. The relatively large difference between the two methods may be attributed to differences in the heating rates used in each technique. However, the exact cause of this discrepancy is still under investigation.

### 5.3. FKM

The FKMs shown in Figure 16 are fluorocarbon-based synthetic rubbers synthesized by copolymerizing vinylidene fluoride (CH_2_=CF_2_), hexafluoropropylene (CF_2_=CFCF_3_), chlorotrifluoroethylene (CF_2_=CFCL), and tetrafluoroethylene (C_2_=F_4_). Due to their excellent chemical stability, thermal resistance, low electrical conductivity, and cost-effectiveness, FKMs are widely employed in various industrial applications [221,222,223,224,225]. However, various studies have reported on the dielectric properties of NBR, SBR, and EPDM copolymers. In contrast, the dielectric dispersion analysis of FKM copolymers remains rare [129,130,131,132,133,134,168,172]. The chemical compositions of the FKM used in this study are summarized in Table 3.

Figure 17a shows three-dimensional dielectric loss spectra over a wide range of temperatures (233 K–373 K) and frequencies (0.01 Hz–3 MHz). Within these temperature and frequency ranges, MWS relaxation, α relaxation, β relaxation, and the DC conduction process were observed. The arrows indicate the evolution of the corresponding relaxation type during temperature variation. Each temperature range used to observe these relaxation phenomena is presented (Figure 17b) as follows:(i).The temperature range for β relaxation: 233 K to 308 K.(ii).The temperature range for α relaxation: 268 K to 308 K.(iii).The temperature range for MWS relaxation: 268 K to 373 K.(iv).The temperature range for the DC conduction contribution: 283 K to 373 K.

To extract valuable physical parameters from the dispersion spectra, the dielectric frequency response data for each temperature point were deconvoluted, as illustrated in Figure 18. The data were fitted via a combination of HN functions and the power law to account for DC conductivity contributions. Four main relaxation peaks were identified, and DC contributions emerged in the high-temperature and low-frequency ranges. Each β and MWS relaxation displays a symmetrical peak shape, and the peak for α is presented as an asymmetrical line shape. Furthermore, the linewidth of MWS relaxation was narrower than that of the α and β relaxations.

The α relaxation in FKM arises from the polarization of the polymer backbone (segmental chain) and is associated with the glass transition dynamics of the polymeric system. In contrast, the β relaxation originates from the rotational fluctuations and local motions of the trifluoromethyl groups (TFM:CF3) in the hexafluoropropylene (HFP) units. The molecular origins and corresponding motional modes of these dielectric relaxation processes are schematically illustrated in Figure 19.

The deconvolution of the dielectric loss spectra allowed for the extraction of the center frequencies (f₀) corresponding to the α, β, and MWS relaxation processes, as well as DC conductivity (σ_dc). These values were plotted against the reciprocal of temperature, as shown in Figure 20. In this plot, both the β and MWS relaxations exhibit a linear dependence on the reciprocal temperature and a logarithmic variation in f_0. Therefore, fitting for β and MWS relaxation was performed via the Arrhenius temperature dependence function (Eq. 2.). From these fitting results, the relaxation rates for the β and MWS relaxation processes provided activation energies of Ea = 50.88 kJ/mol and Ea = 88.57 kJ/mol, respectively. The logarithm of DC conductivity also decreased as the reciprocal of T increased; the Ea of DC conductivity was calculated via the VFTH model (Eq. 5.) as 3.83 eV. These activation energy values are consistent with those reported in four previous studies, as summarized in Table 4. Despite differences in sample preparation and composite conditions, the obtained activation energy values were comparable with previously reported values.

Similarly to a previous study on NBR and EPDM rubber compounds, we evaluated the rate of α relaxation for FKM via the VFTH temperature dependence law. Consequently, the T_g_ value of the FKM was obtained via dielectric relaxation spectroscopy (DRS) and compared with that obtained via DSC measurements. The results obtained using both measurement methods are presented in Figure 21a,b. The T_g_ value obtained via dielectric measurements was 252.6 K, and the T_g_ value obtained via DSC measurements was 253.9 K. Both results revealed comparable T_g_ values with an uncertainty of less than 1 K.

## 6. Conclusions

This study involved a comprehensive analysis of the dielectric responses of nitrile–butadiene rubber (NBR), ethylene–propylene–diene monomer (EPDM), and fluoroelastomer (FKM) copolymers across a wide range of frequencies and temperatures. A self-developed “Dispersion Analysis” program, implemented in C#, was utilized to deconvolute the dielectric relaxation spectra. This program utilized a smoothness function to constrain fitting parameters across multiple temperature ranges, enabling the automated tracking of relaxation peaks with temperature variations. Consequently, this developed program allows for precise and efficient deconvolution analysis via a combination of various fitting functions and related parameters.

The measured dielectric spectra revealed four main relaxation processes, including the DC conduction process in NBR, namely, α, α’, β, and γ relaxation. In contrast, three main α, β, and Maxwell‒Wagner‒Sillars (MWS) relaxations and DC conduction processes were identified in EPDM and FKM. The origins of these observed relaxation processes were identified entirely in relation to the molecular structure of the polymer chain. The activation energies for α’, β, γ, MWS, and DC conductivity were determined by applying the Arrhenius temperature dependence law to the position of the imaginary loss peak versus the reciprocal temperature. The activation energies for NBR and FKM were also compared with previously published results. Furthermore, the glass transition temperatures (T_g_) for three rubber polymers was estimated via Vogel–Fulcher–Tammann–Hesse (VFTH) fitting to the rate of α relaxation. The T_g_ values obtained through dielectric spectroscopy agreed well with the differential scanning calorimetry (DSC) measurements.

Additionally, by analyzing the temperature dependence of the dielectric relaxation strength and the position of the relaxation peak in EPDM, the emergence and splitting of α and β relaxations in EPDM were observed for the first time in this study.

In conclusion, this study provides fundamental insights into the dielectric properties of NBR, EPDM, and FKM, which serve as valuable references for future research and the development of advanced rubber materials.

## Figures and Tables

**Figure 1 polymers-17-01539-f001:**
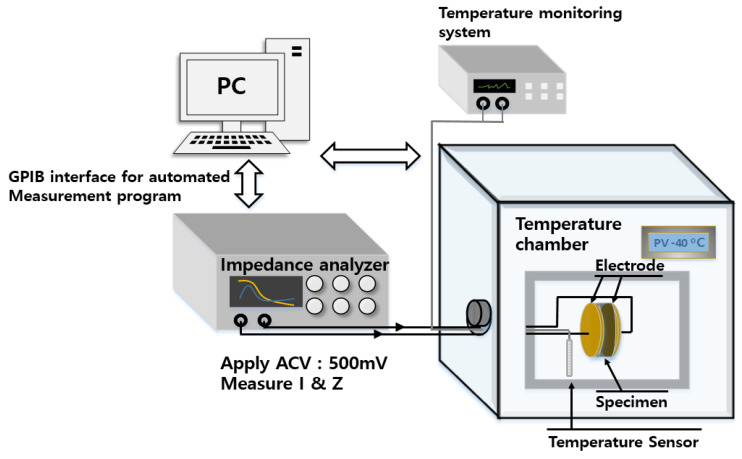
Configuration of automatic dielectric spectroscopy system for measuring complex permittivity in polymer specimens.

**Figure 2 polymers-17-01539-f002:**
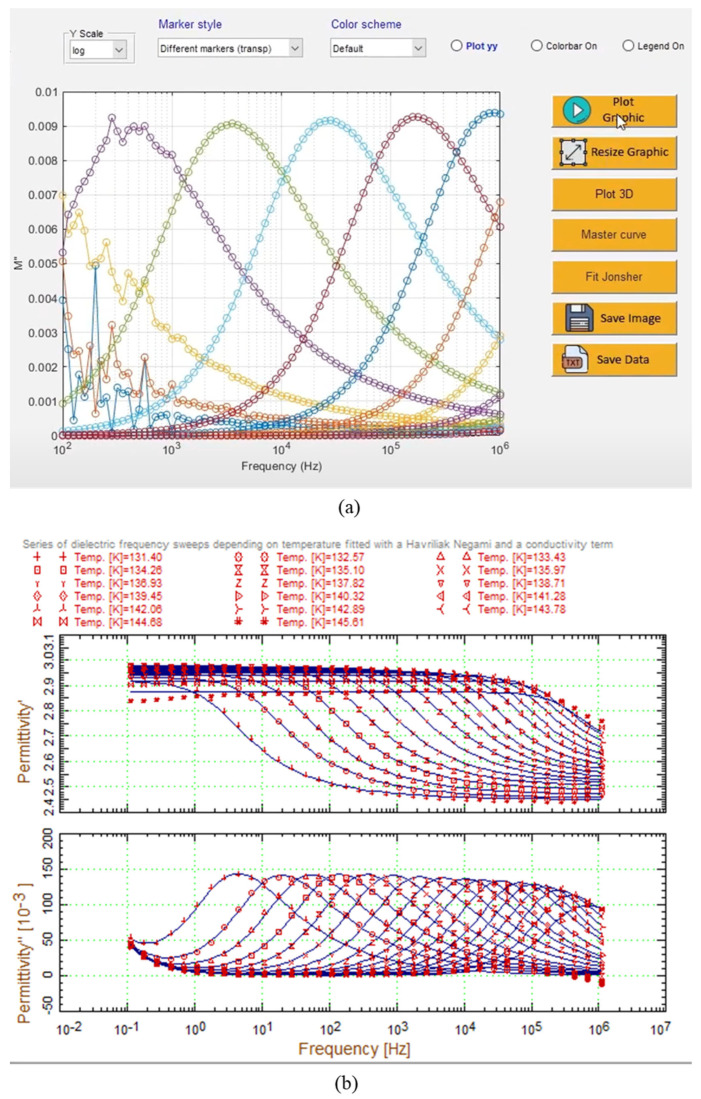
Screenshot images of commercially available data analysis software for determining dielectric dispersion spectra. (**a**) “EIS-smart-tool” (source: https://github.com/bmgmelo/EIS-smart-tool, accessed on 1 February 2025) and the different colors of the empty circles and solid lines indicate the measured data and their respective fitting results at each temperature; (**b**) “WinFit”, Novocontrol Technologies (source: https://www.novocontrol.de/php/winfit.php, accessed on 1 February 2025).

**Figure 3 polymers-17-01539-f003:**
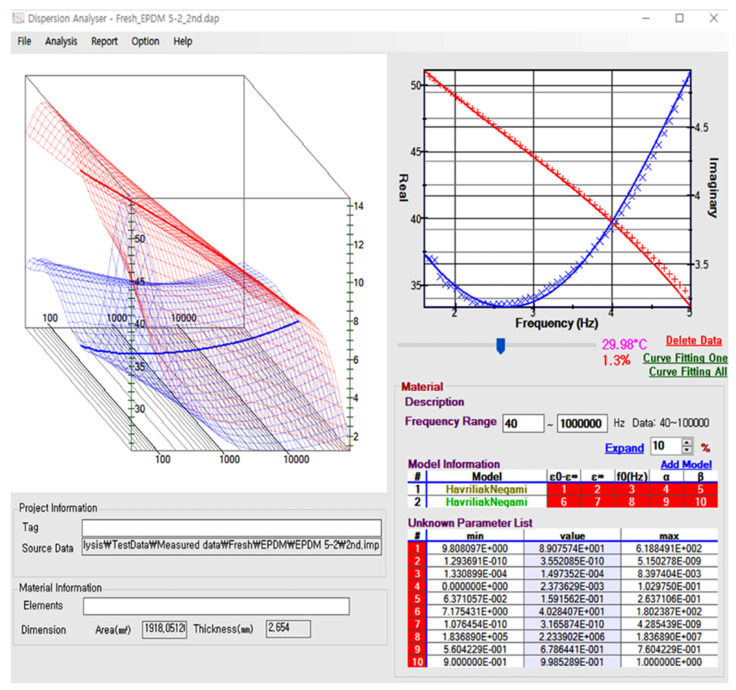
A screenshot showing the execution of the integrated deconvolution analysis program. One typical dataset is loaded as an example, where a 3D graph of complex dielectric constants as a function of frequency and temperature appears in the upper left panel. The red and blue sheets represent the real and imaginary parts of the complex permittivity, respectively. The x- and y-axes represent frequency and temperature, while the double z-axes in Figure 3 represent the real (left) and imaginary (right) permittivity values. In the right half of the main panel, the measured and fitted spectra at a specific temperature are shown. The red and blue cross symbols indicate the real and imaginary values of the measured permittivity, respectively, while the solid lines represent the fitted results for the corresponding colored data using a combination of two HN functions. The ten fitted parameters are listed in the bottom right panel. The fitting accuracy value (1.3%) is shown below the current temperature (29.98 °C), and the details of its calculation procedure are described in the Appendix A.

**Figure 4 polymers-17-01539-f004:**
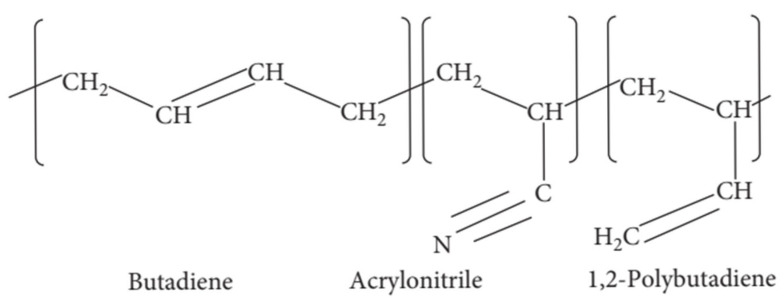
Molecular structure of NBR.

**Figure 5 polymers-17-01539-f005:**
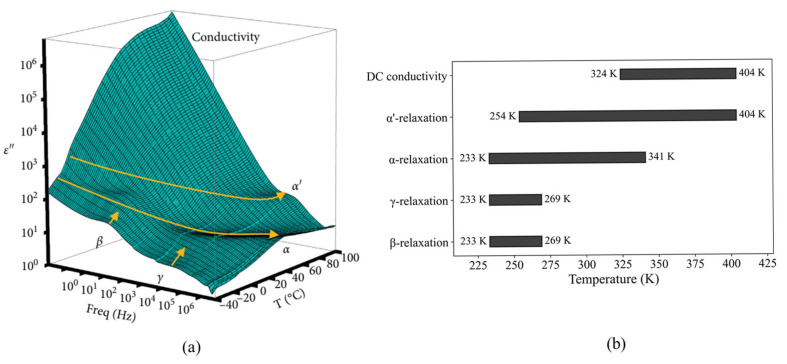
(**a**) A three-dimensional plot of the imaginary permittivity of an NBR; (**b**) the temperature ranges observed for each relaxation process presented as a bar chart.

**Figure 6 polymers-17-01539-f006:**
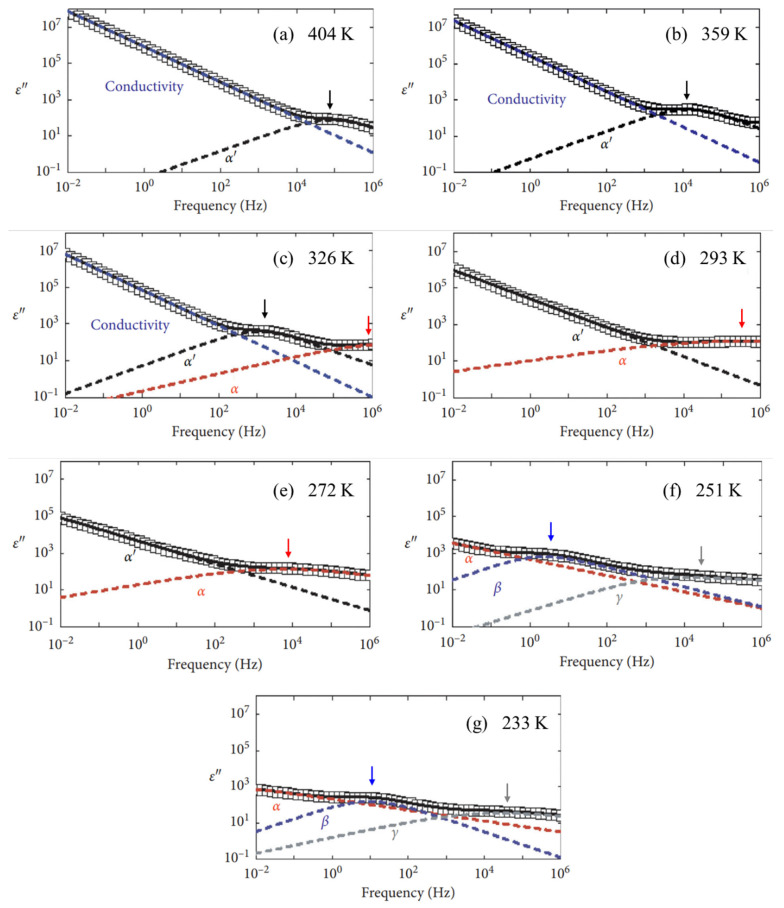
Deconvoluted relaxation process of NBR for seven representative temperatures: (**a**) 404 K, (**b**) 359 K, (**c**) 326 K, (**d**) 293 K, (**e**) 272 K, (**f**) 251 K, and (**g**) 233 K. Contributions made by five relaxation processes—α’ (black dashed line), α (red dashed line), β (blue dashed line), γ (gray dashed line), and conduction (navy dashed line)—to imaginary permittivity data. Black, red, blue, and gray arrows indicate peak position of corresponding α’, α, β, and γ relaxations, respectively.

**Figure 7 polymers-17-01539-f007:**
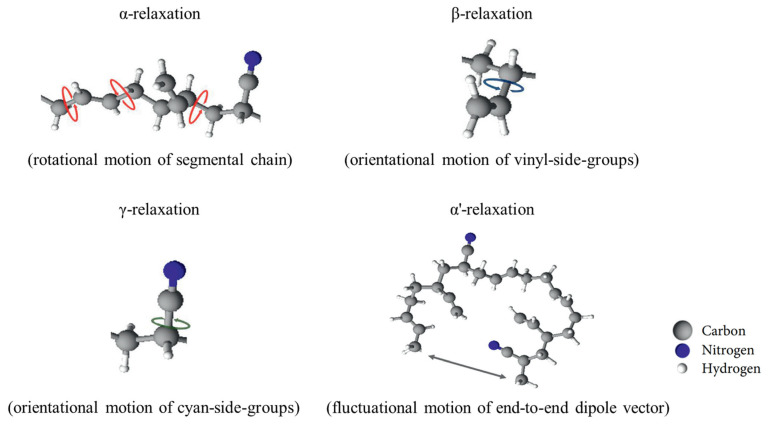
Assignment of dielectric relaxation process for schematic molecular chain structure and its motion mode in NBR.

**Figure 8 polymers-17-01539-f008:**
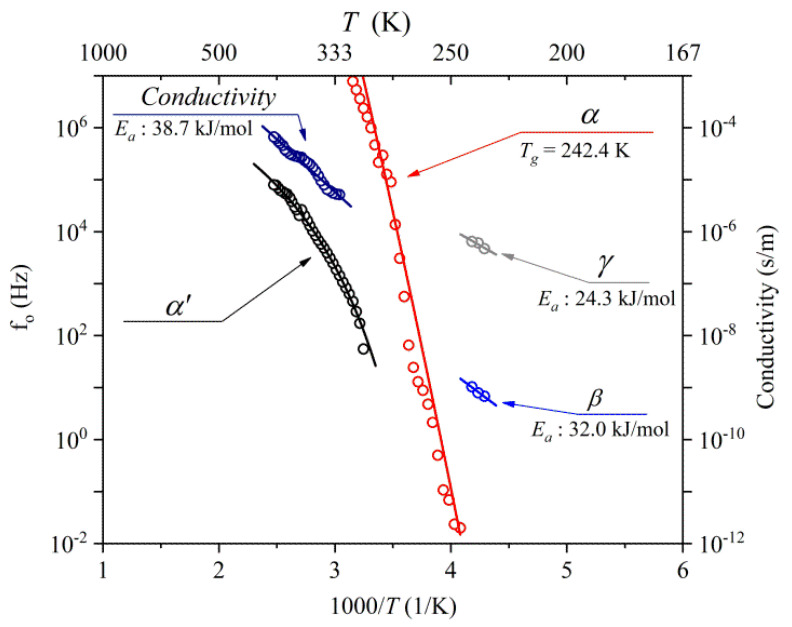
The reciprocal temperature dependence of the central frequency ($f_{0}$) of the imaginary loss peak for the α’ (black circle), α (red circle), β (blue circle), γ (gray circle), and conductivity (navy circle) processes in NBR. Solid lines represent the fitted curves for each relaxation mode. Furthermore, α’ was also fitted using the VFTH model, yielding fitting parameters of $f_{p,\infty }$ = 2.8 × 10^7^ Hz and A = 0.09 eV and a Vogel temperature of 223.239 K.

**Figure 9 polymers-17-01539-f009:**
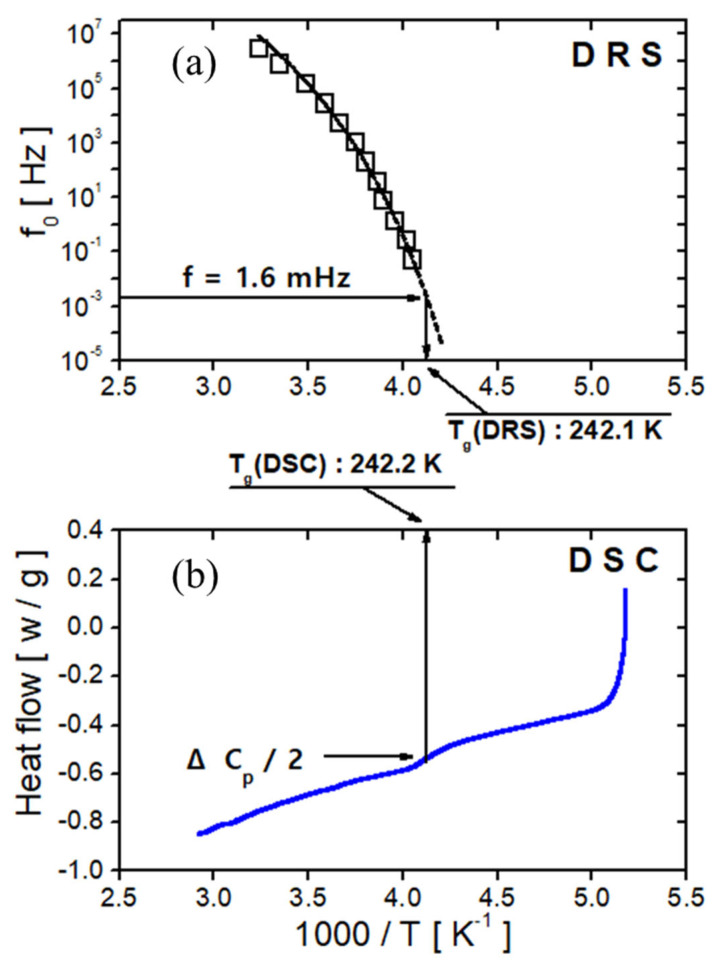
(**a**) The glass transition temperature determined via dielectric spectroscopy, with the dashed line representing the fitted result of the VFTH equation. (**b**) The glass transition temperature determined via DSC.

**Figure 10 polymers-17-01539-f010:**
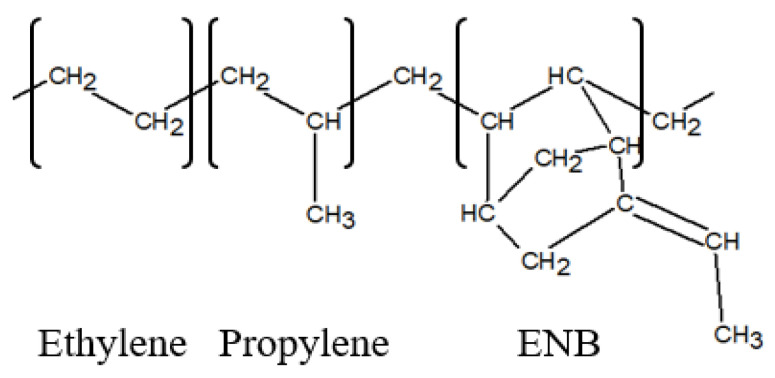
Molecular structure of EPDM.

**Figure 11 polymers-17-01539-f011:**
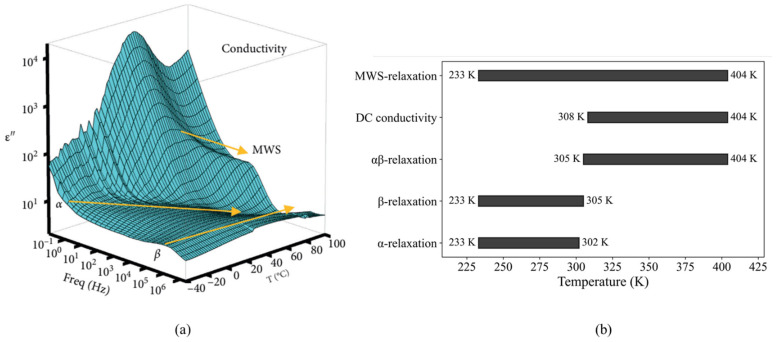
(**a**) A 3D plot of the imaginary part of complex permittivity for EPDM. (**b**) The temperature ranges for the observed relaxation phenomena presented as a bar chart.

**Figure 12 polymers-17-01539-f012:**
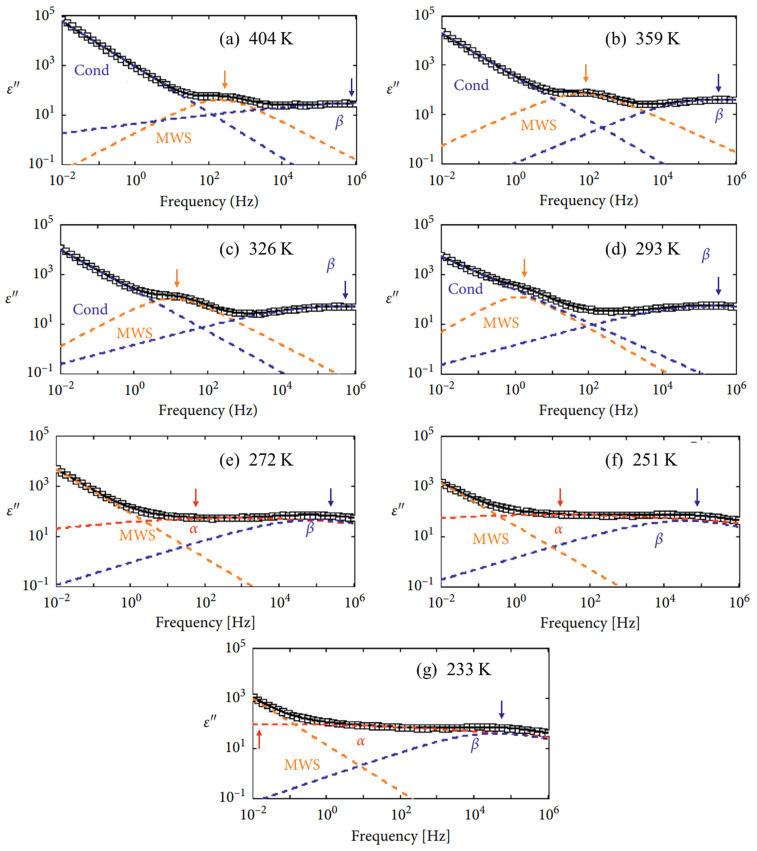
Deconvoluted relaxation process of EPDM for seven representative temperatures: (**a**) 404 K, (**b**) 359 K, (**c**) 326 K, (**d**) 293 K, (**e**) 272 K, (**f**) 251 K, and (**g**) 233 K. Contributions made by all three relaxation processes—MWS (orange dashed line), α (red dashed line), and β (blue dashed line)—and conduction (navy dashed line) to imaginary permittivity data. Orange, red, and blue arrows indicate peak position of corresponding MWS, α, and β relaxations, respectively.

**Figure 13 polymers-17-01539-f013:**
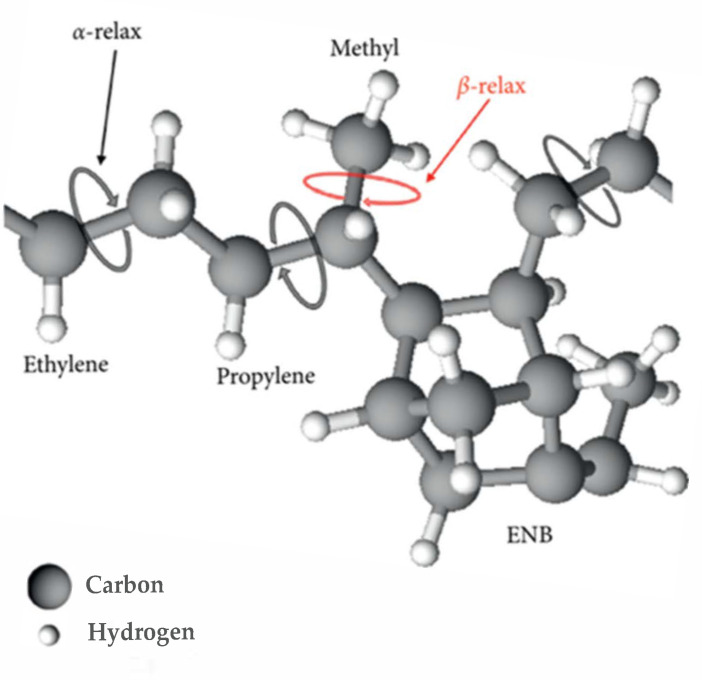
Schematic assignment for molecular structure of EPDM. Each reorientational relaxation is shown with arrows.

**Figure 14 polymers-17-01539-f014:**
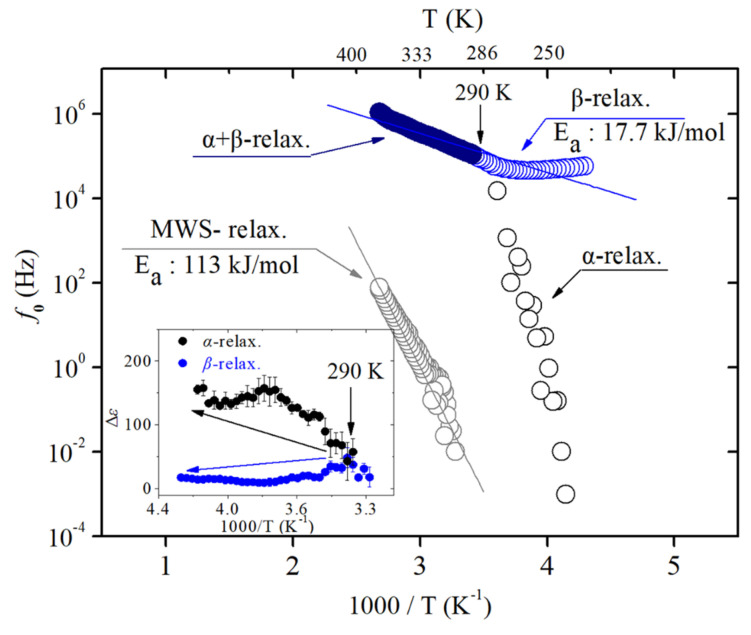
The relaxation rate (f₀) as a function of the reciprocal temperature (1000/T) for EPDM. Activation energies were extracted from the slopes of the fitted solid lines. The overlapping circle symbols for the MWS and α relaxation processes indicate repeated measurements. The inset displays the temperature dependence of the relaxation strength (Δε) for each process.

**Figure 15 polymers-17-01539-f015:**
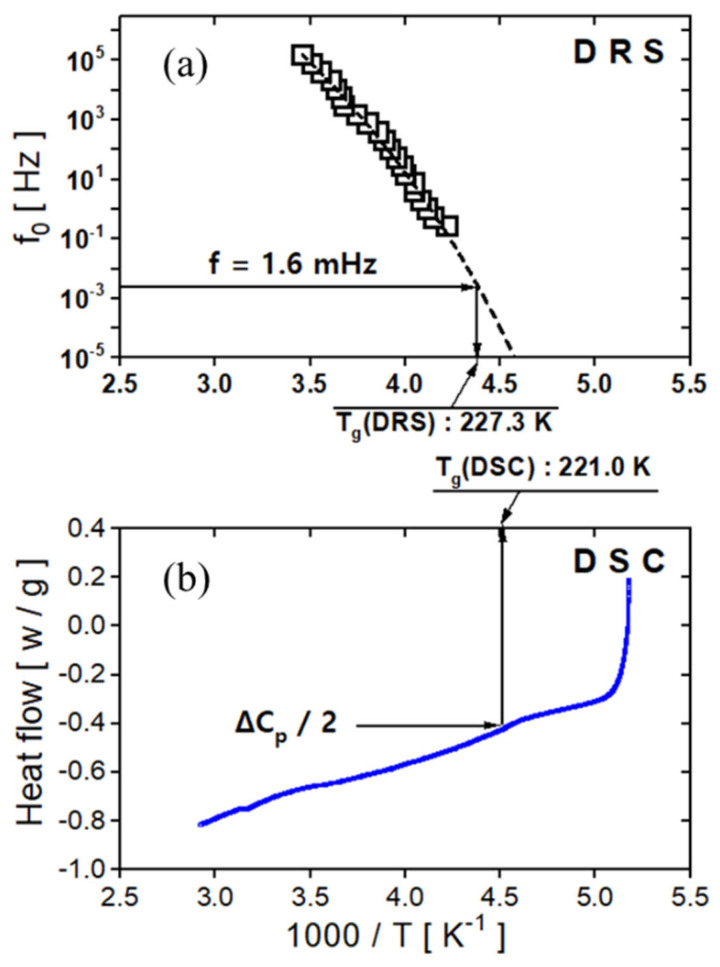
(**a**) The glass transition temperature determined using dielectric spectroscopy, with the dashed line representing the fitted result of the VFTH equation. (**b**) The glass transition temperature determined using DSC.

**Figure 16 polymers-17-01539-f016:**
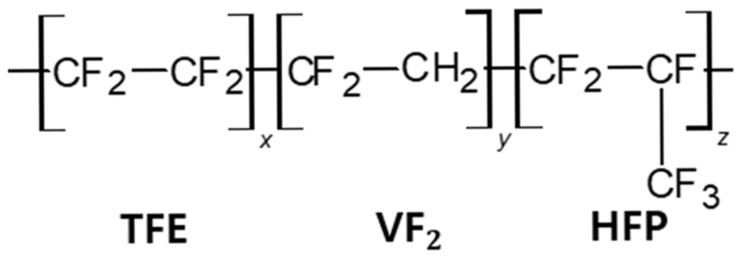
Molecular structure of FKM.

**Figure 17 polymers-17-01539-f017:**
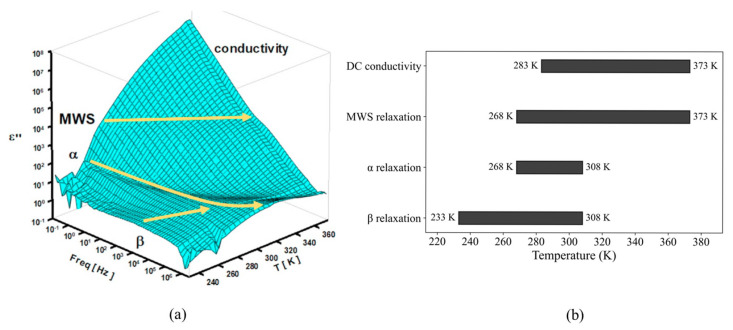
(**a**) A three-dimensional plot of the imaginary permittivity of an FKM. (**b**) The temperature ranges for the observed relaxation phenomena presented as a bar chart.

**Figure 18 polymers-17-01539-f018:**
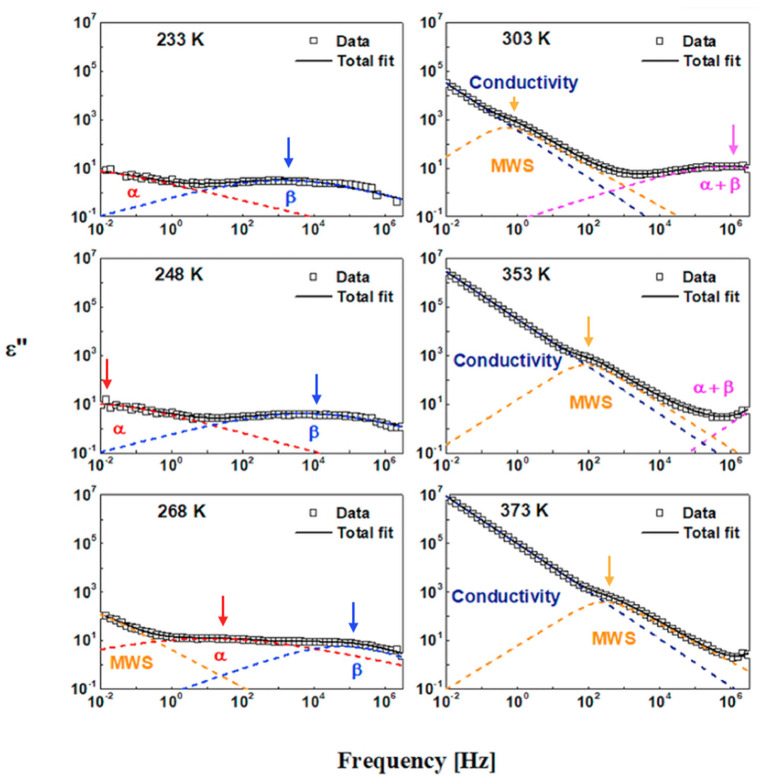
The deconvoluted relaxation process of FKM for several representative temperatures. Both analyzing processes use the combination of HN functions and the power law of the DC conductivity contribution to fit the spectra. All three main relaxations—MWS (orange dashed lines), α (red dashed lines), and β (blue dashed lines)—and conductivity (navy dashed line) were observed in FKM compounds. The arrows indicate the peak positions of the corresponding MWS, α, and β relaxations, respectively.

**Figure 19 polymers-17-01539-f019:**
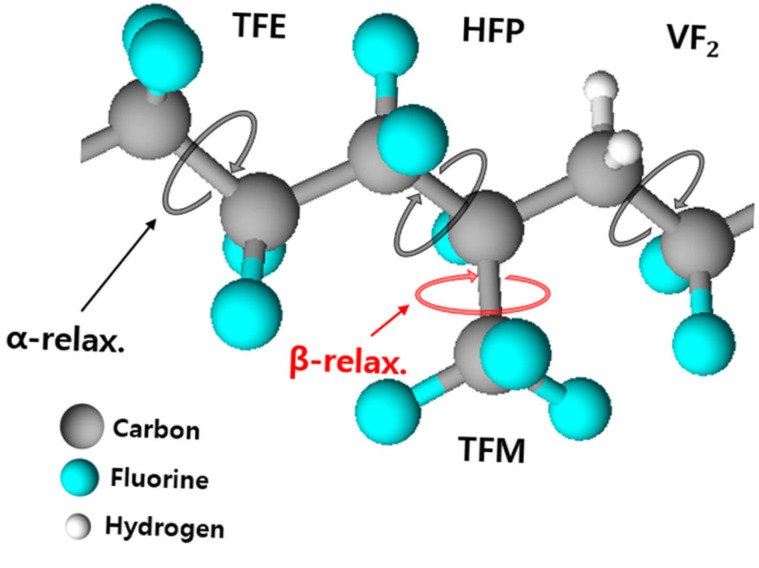
Assignments of the dielectric relaxation processes (α and β) for a schematic molecular chain structure and its motional mode in FKM.

**Figure 20 polymers-17-01539-f020:**
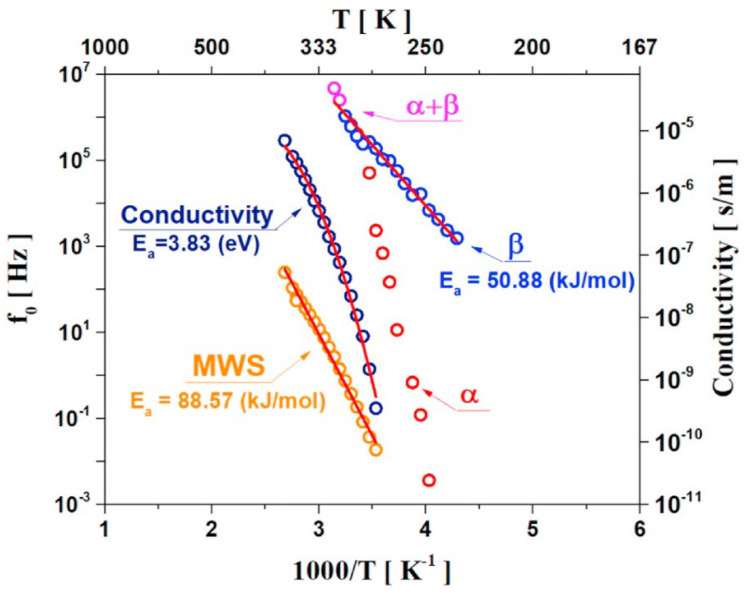
The reciprocal temperature dependence of the central frequency ($f_{0}$) of the imaginary loss peak for the α (red circle), β (blue circle), α + β (pink circle), MWS (orange circle), and conductivity (navy circle) processes in FKM. The activation energy for the relaxation process was the slope of the fitted corresponding solid line.

**Figure 21 polymers-17-01539-f021:**
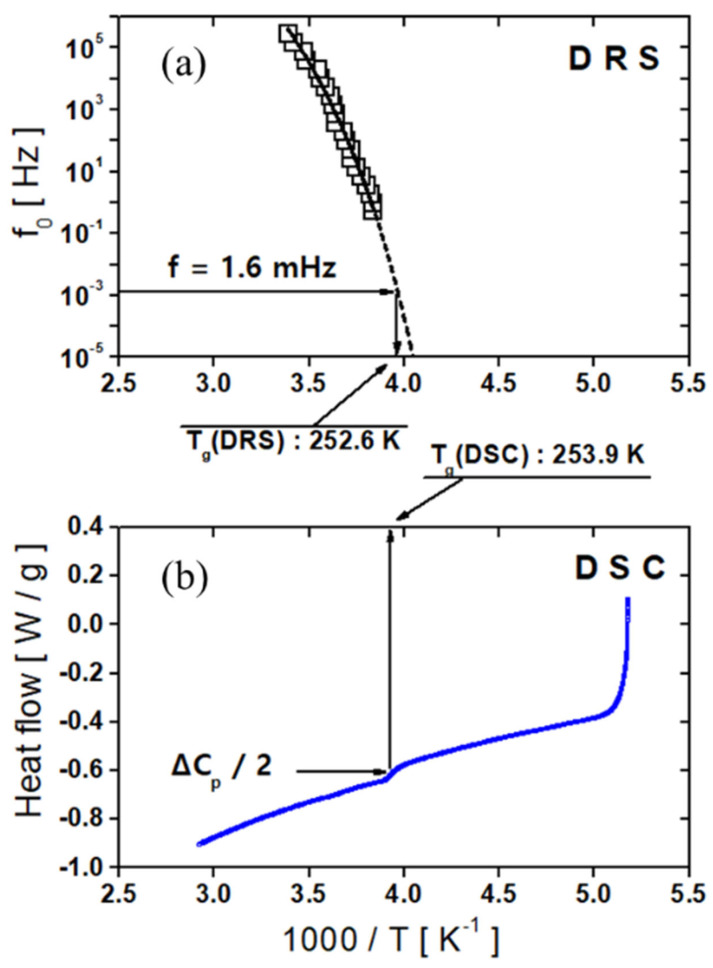
(**a**) The glass transition temperature determined using dielectric spectroscopy, with the dashed line representing the fitted result of the VFTH equation. (**b**) The glass transition temperature determined via DSC.

**Table 1 polymers-17-01539-t001:** The chemical composition of the NBR compound.

Chemical Name	Function	(%)
Acrylonitrile–butadiene rubber	Polymer	40.0
Carbon black (medium thermal)	Filler reinforcing	50.0
1,2-Benzenedicarboxylic acid	Processing aid	6.0
2-Benzimidazolethiol	Antioxidant	2.0
Sulfur	Curing agent	2.0
Total		100.0

**Table 2 polymers-17-01539-t002:** Chemical composition of EPDM rubber compound.

Chemical Name	Function	(%)
Ethylene propylene rubber	Polymer	58.0
Carbon black	Filler reinforcement	34.0
Zinc oxide	Processing aid	3.0
Dicumyl peroxide	Antioxidant	5.0
Total		100.0

**Table 3 polymers-17-01539-t003:** Chemical composition of FKM rubber compound.

Chemical Name	Function	(%)
Poly (vinylidene fluoride-co-hexafluoropropylene)	Polymer	82.0
Carbon black	Filler reinforcement	14.0
Calcium dihydroxide	Curing agent	4.0
Total		100.0

**Table 4 polymers-17-01539-t004:** Comparison of activation energy values for β and MWS relaxations in FKM measured using dielectric spectroscopy.

Reference	Ours	[172]	[226]	[133]	[227]
E_a_ for β (kJ/mol)	50.88	63.5	50.2, 54.4 *	47	49.8, 40.2
E_a_ for MWS (kJ/mol)	88.57	110		151, 81 **	

* Referred to as γ relaxation. ** Referred to as α relaxation.

## Data Availability

The data used to support the findings of this study are available from the corresponding author upon request.

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
