# Peer review of "A Study of the Dielectric Relaxation of Nitrile–Butadiene Rubber, Ethylene–Propylene–Diene Monomer, and Fluoroelastomer Polymers with a Self-Developed Deconvolution Analysis Program"

_polymers, 2025, doi:10.3390/polym17111539_

Round 1
Reviewer 1 Report
Comments and Suggestions for Authors
This study reports a useful technique to fit dielectric relaxation spectra and obtain multiple relaxation peaks in a self-consistent, reliable manner. The fitting techniques and the obtained results are worthy of publication but the manuscript might need more changes
The motivation of the study needs to be described better - as per the title, this study is about rubbers, then in the introduction, the authors should address the reasons why dielectric properties have been poorly described for this material so far and what is the need for further detailed studies such as the manuscript under consideration.
The novelty of the study is the fitting. Overall, the authors need to highlight the unique performance capabilities of their method better as well as explain how the user can associate errors to the fit.
Accept with minor revisions
Suggestions to authors
1. Choose one material system and compare the performance of the developed method to winfit or eis-smart-tool
2. Page 3, Line 106 cite reference for 1.6mHz
3. Page 5, Line 176 clarify what approaches 0
4. What are the metrics of quality of the final fit? How is the final fit approved? Describe error reporting and quality control of the fit in detail.
5. Why is MWS present in the second/third case and why not in the first? It arises out of interfacial polarization- what is the interface here? And in the first case, what happens when particle-polymer interfaces occur? Why has electrode polarization not been considered in second and third material systems?
6. Does the ferroelectric nature of PVDF affect the measurement of the elastomer?
7. Sample preparation - polymerization and crosslinking. What happens to the uncrosslinked chains? How do they impact the fitting? Are these samples pliant enough to make good contact with the electrode surface, especially below Tg? What is the degree of crosslinking? Describe these details in the manuscript.
Author Response
Dear editor and reviewers,
We are grateful for your thoughtful comments on our submitted manuscript, and we appreciate the time and effort you have put into reviewing our work. Your feedback has been invaluable in helping us to improve the quality of our research. Please find our responses to your comments below. Each response is provided below, corresponding to the reviewer's comments.
Response to Reviewer #1’s comments
This study reports a useful technique to fit dielectric relaxation spectra and obtain multiple relaxation peaks in a self-consistent, reliable manner. The fitting techniques and the obtained results are worthy of publication but the manuscript might need more changes.
The motivation of the study needs to be described better - as per the title, this study is about rubbers, then in the introduction, the authors should address the reasons why dielectric properties have been poorly described for this material so far and what is the need for further detailed studies such as the manuscript under consideration.
The novelty of the study is the fitting. Overall, the authors need to highlight the unique performance capabilities of their method better as well as explain how the user can associate errors to the fit.
Accept with minor revisions
Reviewer’s comment 1:
Choose one material system and compare the performance of the developed method to winfit or eis-smart-tool
Our answer: We would like to express our sincere appreciation to Reviewer #1 for the meticulous evaluation and constructive comments on our manuscript. Their insightful suggestions have significantly contributed to improve the clarity and depth of our work.
As noted, we understand the importance of benchmarking our developed method against existing tools such as WinFit and EIS-Smart Tool. Unfortunately, we were unable to perform a comparison with WinFit due to licensing limitations, as it is a commercial software package that we do not currently have access.
Alternatively, we attempted to evaluate our method using the EIS-Smart Tool. While the software offers a wide range of fitting and modeling functions, it does not provide a clear or easily accessible feature for quantitatively assessing fitting accuracy—such as statistical goodness-of-fit metrics or residual analysis tools—which made a direct comparison with our analysis program difficult.
Furthermore, out of respect for the developers of commercial tools, we refrained from including any direct performance critique in the manuscript. Instead, we have revised the text to clearly emphasize the unique advantages and technical strengths of our in-house developed tool. In the revised manuscript (line 231-251, marked in red), we have added the following description to highlight these points:
“However, in practical experimental settings, impedance data are acquired across numerous temperature points to track the evolution of dielectric relaxations with thermal variation. As a result, these tools require repetitive manual fitting procedures for each da-taset, significantly increasing the analysis time and reducing overall efficiency.
To address this challenge, we developed a custom data analysis software named “Dispersion Analyzer” to facilitate comprehensive and consistent deconvolution across wide temperature ranges. The software incorporates a smoothness constraint function that enables simultaneous fitting of multiple datasets while maintaining physical conti-nuity between temperature-dependent parameters. Unlike conventional tools that fit each temperature individually, our software supports:
- Weighted fitting between the real and imaginary parts of permittivity spectra to emphasize dominant spectral contributions,
- Global fitting across temperature series under physically meaningful constraints (e.g., shared activation energy or Arrhenius/VFT behavior),
- User-controllable fitting accuracy by adjusting convergence thresholds, iteration limits, and optimization tolerances.
These capabilities allow for both interpretability and efficiency in complex dielectric analysis, which—based on our assessment—are not commonly supported in existing commercial packages.
The developed software and additional information are provided in the Supplemen-tary Materials.”
These features collectively provide a highly flexible and physically interpretable framework for dielectric spectrum analysis, which—based on our knowledge—is not commonly available in existing commercial tools.
We kindly ask for the reviewer’s understanding on this matter, and we hope that the revised manuscript and expanded discussion on our method’s capabilities adequately address the reviewer’s comment.
Reviewer’s comment 2:
Page 3, Line 106 cite reference for 1.6mHz
Our answer: As suggested by the reviewer, the reference for the determination of glass transition temperature by dielectric spectroscopy has been added and marked in red (Line 106).
Reviewer’s comment 3:
Page 5, Line 176 clarify what approaches 0
Our answer: We appreciate the reviewer’s valuable comment and apologize for the lack of clarity in the original expression. In Page 5, Line 176, we intended to describe the evolution of the relaxation spectrum with respect to the parameters α and β in the Havriliak–Negami equation. However, the phrasing was ambiguous. In response, we have revised the sentence to: “As α and β approach zero, the relaxation spectrum exhibits increased broadening and a more asymmetrical shape.”.
Reviewer’s comment 4:
What are the metrics of quality of the final fit? How is the final fit approved? Describe error reporting and quality control of the fit in detail.
Our answer: In our work, the quality of the final fit is quantitatively assessed using a Figure of Merit (FOM), which is defined as the sum of squared deviations between the measured and calculated complex permittivity data:
Where
,
This error metric is implemented flexibly to support different types of discrepancy measures:
Linear:
Logarithmic:
where δ is an offset added to avoid divergence near zero.
Mixed: combination of linear and logarithmic terms.
The final fit is approved based on one or more of the following criteria:
- The FOM falls below a user-defined threshold.
- The optimization algorithm (e.g., Simplex method) converges within a maximum number of iterations.
- All fitted parameters remain within physically meaningful and predefined bounds.
Error reporting is carried out by displaying the FOM values in the main and fitting status graphical interface. Furthermore, we provide residual plots of real and imaginary components to allow visual inspection of the fitting quality. These combined procedures constitute our quality control framework, ensuring both quantitative accuracy and qualitative reliability of the fit. Relevant details have been added to Section A of the Supplementary Information.
Reviewer’s comment 5:
Why is MWS present in the second/third case and why not in the first? It arises out of interfacial polarization- what is the interface here? And in the first case, what happens when particle-polymer interfaces occur? Why has electrode polarization not been considered in second and third material systems?
Our answer: As the reviewer correctly points out, MWS relaxation typically originates from interfacial polarization between phases with strong dielectric contrast, such as the interface between a polymer matrix and conductive or high-permittivity fillers. While the NBR system also contains carbon black fillers, we did not observe a distinguishable MWS relaxation feature in the dielectric spectra.
This absence is consistent with our prior studies on NBR systems with varying filler content, where even above the electrical percolation threshold, no significant MWS relaxation was identified [Kim et al., Polymers, 14(1), 155, 2022].
Based on these findings, we expected that NBR may exhibit weaker interfacial polarization or a smaller permittivity contrast between the matrix and the filler, which could suppress the emergence of MWS relaxation. However, we emphasize that this remains a working hypothesis, and further structural and electrical characterization will be necessary to confirm the underlying this mechanism. Therefore, we plan to explore this systematically in future studies.
Reviewer’s comment 6:
Does the ferroelectric nature of PVDF affect the measurement of the elastomer?
Our answer: It is well known that polyvinylidene fluoride (PVDF) can exhibit ferroelectric behavior, which may contribute to additional polarization processes in dielectric spectroscopy measurements. Such effects can potentially enhance the observed dielectric permittivity and introduce complex relaxation features, especially at low frequencies.
In our study, we did not perform a quantitative analysis of the PVDF content nor specifically characterize the ferroelectric contribution. However, it is reasonable to consider that the ferroelectric nature of PVDF might have influenced the dielectric response of the composite system to some extent. This possibility is supported by prior reports, such as Zhang et al. [Materials Today: Proceedings, 74, 293–297, 2023], which demonstrate that PVDF-containing fluoroelastomers may exhibit enhanced dielectric properties due to intrinsic polarization behavior.
While this contribution was not separately analyzed in the present study, we acknowledge its potential impact and will consider further investigations in future work to better understand the role of ferroelectric components in the dielectric behavior of elastomeric systems.
Reviewer’s comment 7:
Sample preparation - polymerization and crosslinking. What happens to the uncrosslinked chains? How do they impact the fitting? Are these samples pliant enough to make good contact with the electrode surface, especially below Tg? What is the degree of crosslinking? Describe these details in the manuscript.
Our answer: Crosslink density in polymer refers to the number of chemical bonds formed between polymer chains, creating a three-dimensional network. It directly influences the material's mechanical and physical properties, such as stiffness, strength, and thermal stability. Crosslinking mostly takes place via vulcanization with sulfur but is also accomplished with peroxides for improved heat resistance. The degree of crosslink density was determined by equilibrium swelling. The swelling experiments were conducted by immersing the sample in terahydrofuran (THF) at ambient temperature for 72 hours. The weight of rubber sample before and after the immersion was measured. The crosslink density (ν) of units in (mol/g) for the polymer composites is calculated according to Flory-Rehner equations [1-3]. The crosslink densities for NBR and EPDM used in this study were measured to be 7.39 and 32.2 x 10-5 mol/g. However, the density for FKM was not measured at that time.
Because of the time constraints during this revision process, it was difficult to measure the crosslinking again or to perform a quantitative investigation on the specific effects of crosslinking on the dielectric relaxation processes. Therefore, although it is a noteworthy comment, we unfortunately decided not to include discussions about crosslinking effects in the manuscript. We plan to address this aspect more thoroughly in future studies.
[1] Lee, J. Y.;Park, N.;Lim, S.; Ahn, B.; Kim, W.; Moon, H.; Paik, H. J.;Kim, W. Influence of the silanes on the crosslink density and crosslink structure of silica-filled solution styrene butadiene rubber compounds. Composite Interfaces 2017, 1267524; DOI:10.1080/09276440.2017.1267524.
[2] Marzocca, A.J. Evaluation of the polymer–solvent interaction parameter for the system cured styrene butadiene rubber and toluene. European Polymer Journal 2007, 43, 2682–2689; DOI:10.1016/j.eurpolymj.2007.02.034.
[3] Hrnjak-Murgic, Z.; Jelencic, J.; Bravar, M.; Marovic, M. Influence of the network on the interaction parameter in system EPDM vulcanizate-solvent. J. Appl. Polym. Sci. 1998, 65, 991-999; DOI:10.1002/(SICI)1097-4628(19970801).
Please let me know if there are still additional comments or insufficient explanation/discussion.
Thanks again for invaluable comments.
With best wishes,
Jae Kap Jung

Reviewer 2 Report
Comments and Suggestions for Authors
This paper gives a special comment about the dielectric relaxation of nitrile–butadiene rubber, ethylene–propylene–diene monomer and fluoroelastomer polymers with a self-developed deconvolution analysis program developed by authors. The self-developed deconvolution analysis program is useful. I recommend it to publish after considering the following the comments.
- The temperature σdc dependence of σdc can be described by the VFTH model in Eq. 5 may be incomplete because electron carrier often follows Arrhenius temperature dependence and ions sometimes follow Arrhenius temperature dependence in a glassy matrix.
- 10 gives a HN function formalism containing a combination of the conductivity term with several relaxation terms. How to treat electrode polarization? It is included in several relaxation terms? Please explain it in order to merit readers.
- In fig. 7, the beta relaxation process is not clear and covered by large dc conductivity. It is better to use the derivative formalism of the real part of permittivity based on the Kramers-Kronig relations to assist our data analysis to clarify the beta relaxation characteristic.
- In fig9, the activation energy of α’relaxation (black circle) by Arrhenius temperature dependence is suitable? It seems to follow VFT equation.
- In fig. 13 and 19, the Alfa relaxation process is not clear and covered by large dc conductivity. It is better to use the derivative formalism of the real part of permittivity based on the Kramers-Kronig relations to assist our data analysis to clarify the Alfa relaxation characteristic. In addition, how to clarify the difference between MWS relaxation and Alfa relaxation? Please give a comment about this also according to the temperature dependence of dielectric intensity. Also, the electrode polarization should be analyzed. It is better to give the real part of permittivity in order to merit readers.
Author Response
Dear editor and reviewers,
We are grateful for your thoughtful comments on our submitted manuscript, and we appreciate the time and effort you have put into reviewing our work. Your feedback has been invaluable in helping us to improve the quality of our research. Please find our responses to your comments below. Each response is provided below, corresponding to the reviewer's comments.
Response to Reviewer #2’s comments
This paper gives a special comment about the dielectric relaxation of nitrile–butadiene rubber, ethylene–propylene–diene monomer and fluoroelastomer polymers with a self-developed deconvolution analysis program developed by authors. The self-developed deconvolution analysis program is useful. I recommend it to publish after considering the following the comments.
Reviewer’s comment 1:
The temperature σdc dependence of σdc can be described by the VFTH model in Eq. 5 may be incomplete because electron carrier often follows Arrhenius temperature dependence and ions sometimes follow Arrhenius temperature dependence in a glassy matrix.
Our answer: We are grateful to Reviewer #2 for their careful review of our manuscript. We agree that the temperature dependence of is more appropriately described by an Arrhenius-type behavior, particularly since electron and ion transport in disordered or glassy systems often follows Arrhenius rather than VFTH kinetics. As the reviewer pointed out, our analysis showed that NBR exhibits a nearly linear trend in the Arrhenius plot, indicating thermally activated hopping-type behavior. In contrast, the FKM samples displayed a characteristic VFTH-type curvature in the σ_dc versus reciprocal temperature plot, suggesting the presence of stronger glassy dynamics.
In response to the reviewer’s comment, we have revised the manuscript accordingly and added the following sentence (highlighted in red) in line 154-157:
“In some systems, especially those without strong glassy dynamics, the temperature dependence of is better described by an Arrhenius law. This behavior reflects thermally activated hopping transport through a more uniform energy landscape [200].”
Reviewer’s comment 2:
Eq. (10) gives a HN function formalism containing a combination of the conductivity term with several relaxation terms. How to treat electrode polarization? It is included in several relaxation terms? Please explain it in order to merit readers.
Our answer: We agree with the reviewer’s comment that evaluating the effect of electrode polarization is essential for a comprehensive analysis of the dielectric response in various polymeric materials. In our fitting procedure, the contribution of electrode polarization was accounted for by incorporating the power-law exponent N in the conductivity term of the Havriliak–Negami (HN) formalism, as described in Eq. (10) of the manuscript.
In addition, we clarify that the contribution of electrode polarization was not incorporated into the individual relaxation terms but was exclusively treated through the conductivity term using the exponent N.
As suggested, we have included the temperature-dependent behavior of the exponent N for the EPDM and FKM specimens, along with relevant discussion, in supplementary information (Fig. S3.). A corresponding statement has also been added to the manuscript and marked in red 9 (line 183-187):
“N is an exponent that characterizes the nature of the conduction process as well as the electrode polarization effect on the dielectric dispersion spectrum. This contribution becomes most significant at high temperatures and low frequencies. The temperature dependence of the exponent N for EPDM and FKM is presented in Fig. S1.”
Reviewer’s comment 3:
In fig. 7, the beta relaxation process is not clear and covered by large dc conductivity. It is better to use the derivative formalism of the real part of permittivity based on the Kramers-Kronig relations to assist our data analysis to clarify the beta relaxation characteristic.
Our answer: The current analysis is based on the HN function, which is analytically constructed to satisfy the Kramers–Kronig (KK) relations. While we fully acknowledge the value of an explicit KK consistency check, implementing such a procedure—both in terms of numerical transformation and physical interpretation—requires careful reconfiguration of the analysis code and potentially significant preprocessing of the experimental data. Due to time constraints and the scope of the current study, we have decided to focus on the HN-based approach, which already provides a physically meaningful and self-consistent description of the dielectric response.
We sincerely appreciate the reviewer’s insightful suggestion, and we plan to incorporate KK-based validation in a future extension of this work.
Reviewer’s comment 4:
In fig9, the activation energy of α’relaxation (black circle) by Arrhenius temperature dependence is suitable? It seems to follow VFT equation.
Our answer: Yes, the temperature dependence of the α′-relaxation follows the VFTH behavior. In fact, applying the VFT model to describe the normal mode relaxation is more appropriate and widely accepted than using the Arrhenius law. Therefore, we have revised Figure 9 accordingly to reflect the VFT fitting.
Reviewer’s comment 5:
In fig. 13 and 19, the Alfa relaxation process is not clear and covered by large dc conductivity. It is better to use the derivative formalism of the real part of permittivity based on the Kramers-Kronig relations to assist our data analysis to clarify the Alfa relaxation characteristic. In addition, how to clarify the difference between MWS relaxation and Alfa relaxation? Please give a comment about this also according to the temperature dependence of dielectric intensity. Also, the electrode polarization should be analyzed. It is better to give the real part of permittivity in order to merit readers.
Our answer: We appreciate the reviewer’s valuable suggestion. Below we address each point in detail.
- Clarification of the α-relaxation and KK-based analysis
As mentioned in our response in Comment 3, we plan to incorporate KK-based validation and derivative analysis in a future extension of this work.
- Distinguishing MWS and α-relaxation
We agree that distinguishing between MWS relaxation and α-relaxation is essential. In general, these processes can be characterized by differences in their temperature dependence behavior of relaxation rate. Most of MWS and α-relaxation follows the Arrhenius and VFTH temperature dependence law, respectively. We are grateful to the reviewer’s insightful suggestion, through which we have come to recognize notable features in the dielectric strength (Δε) of MWS and α-relaxation. MWS relaxation tends to exhibit significantly higher Δε due to interfacial polarization, whereas α-relaxation originates from segmental motion near the glass transition. However, in our system, these trends do not consistently apply, and since we are not confident in providing a definitive explanation for this behavior, we chose not to include detailed Δε-based discussions in the manuscript to avoid potential confusion. Nevertheless, related statements regarding the distinction between these relaxation processes are provided in the main text.
- Electrode polarization
Regarding electrode polarization, we included its contribution by modeling the low-frequency dispersion using a conductivity term with a power-law exponent "N" as shown in Eq. (10) of the manuscript. The temperature-dependent behavior of this exponent for EPDM and FKM specimens was provided in the supplementary figures. Interestingly, the N-values for these materials show opposite temperature dependence, indicating complex behavior that warrants further investigation. We appreciate the reviewer’s insight and plan to further address this in future studies.
Additionally, as requested, the real part of the complex permittivity has been added to the supplementary resources (Fig. S2) to provide a more complete view for readers.
We would like to appreciate for both reviewer’s thoughtful evaluations, and it were greatly helpful in improving the quality of our work. We have endeavored to address all points raised by the reviewers, and believe that this version represents a significant improvement, which we submit for your consideration. We look forward to hearing from you.
Please let me know if there are still additional comments or insufficient explanation/discussion.
Thanks again for invaluable comments.
With best wishes,
Jae Kap Jung

Round 2
Reviewer 2 Report
Comments and Suggestions for Authors
The manuscript has been well revised and it can be published now.
Author Response
We sincerely thank Reviewer 2 for the positive evaluation and kind comments. We truly appreciate your time and support throughout the review process.